



# NOAA Air Resources Laboratory Atmospheric Turbulence and Diffusion Division's Measurements of Temperature, Humidity and Wind using Small Uncrewed Aircraft Systems to Support Short-Term Weather Forecasting Needs over Complex Terrain

Temple R. Lee[1], Travis J. Schuyler[1,2], Michael Buban[1,2], Edward Dumas[1,3], Tilden P. Meyers[1], and C. Bruce Baker[1,*]

[1]NOAA Air Resources Laboratory Atmospheric Turbulence and Diffusion Division, Oak Ridge, Tennessee, USA
[2]Cooperative Institute for Severe and High-Impact Weather Research and Operations (CIWRO), Norman, Oklahoma, USA
[3]Oak Ridge Associated Universities, Oak Ridge, Tennessee, USA
[*]Retired as of 1 Jan 2022

*Correspondence to*: Temple R. Lee (temple.lee@noaa.gov)

**Abstract.** Small uncrewed aircraft systems (sUxS) are now being routinely used not only for sampling atmospheric boundary layer (ABL) processes and land-atmosphere interactions but also have significant potential to improve weather forecasting at National Weather Service (NWS) Weather Forecast Offices (WFOs). In the present study, we used
observations obtained from a Meteomatics Meteodrone SSE sUxS flown on 31 days between 20 August and 10 December 2020 near Oliver Springs, Tennessee, located 35 km northwest of Knoxville, Tennessee. We flew the sUxS up to 700 m above ground level, starting around sunrise and continuing every half hour until 3.5-4.0 hours past sunrise under synoptically quiescent, fair weather conditions. These datasets were provided in real time to the local NWS WFO in Morristown, Tennessee and used by forecasters there to assist with short-term operational forecasting needs. The sUxS profiles also
provided finescale details on the early-morning transition over complex terrain and how this evolution varied during the late summer to winter period, which can be used to support the initialization of numerical weather prediction models.

## 1 Introduction

Small uncrewed aircraft systems (sUxS) have been used for measuring atmospheric thermodynamic and kinematic quantities for over two decades and have been shown to successfully close a significant observation gap in the sampling of Earth's
atmosphere (e.g., Holland et al. 2001; Houston et al. 2012; Elston et al. 2015; Palomaki et al. 2017). Many commercially-available sUxS have the ability to operate up to 1-3 km above ground level (AGL), based upon manufacturer-stated specifications. For this reason, measurements obtained from sUxS flights provide a wealth of information on processes in the lowest part of the atmosphere including but not limited to land-atmosphere feedbacks during solar eclipses (e.g., Lee et al. 2018, Bailey et al. 2019), the spatiotemporal variability in surface and near-surface temperature and moisture fields during
the evolution of the daytime atmospheric boundary layer (ABL) (e.g., Lee et al. 2019; Desai et al. 2021; Prior et al. 2021),



valley drainage flows (e.g., Bailey et al. 2020, de Boer et al. 2021), and mesoscale processes within the pre-convective boundary layer (e.g., Koch et al. 2018). Furthermore, sUxS flights during the early-morning transition (Wildmann et al. 2015) and early-evening transition phases of the ABL (e.g., Bonin et al. 2013, Båserud et al. 2020) provide more information on ABL evolution during these periods than can be obtained from surface observations alone.

So far, many studies have focused on sUxS observations over a selection of case studies and short-term deployments during field campaigns (e.g., Koch et al. 2018, Lee et al. 2019, Bailey et al. 2020, Islam et al. 2021, de Boer et al. 2021), but few studies have lasted more than ~ 1 month. We argue having routine, real-time information on the evolution of the ABL has the strong potential to benefit operational weather forecasting needs (Houston et al. 2021; Pinto et al. 2021). In regions where routine rawinsonde observations are unavailable and / or in regions of complex terrain where rawinsonde observations

may not be regionally representative, routine sUxS observations can be made available to National Weather Service (NWS) Weather Forecast Offices (WFO) in near real time to provide forecasters there with current information on the temperature, moisture, and wind fields within the ABL.

To this end, in the present study we conducted 241 sUxS flights over 31 days between 20 August and 10 December 2020 in the complex terrain northwest of Knoxville, Tennessee and provided the datasets obtained from these sUxS flights, which

included measurements of temperature, humidity, pressure, wind speed, and wind direction, to the local WFO in Morristown, Tennessee. To the best of the authors' knowledge, the present study represents the first time that sUxS observations were provided in real-time to a NWS WFO over a multi-month period to assist with operational weather forecasting needs. Additionally, the sUxS observations that were obtained to support this effort can provide insights into the evolution of the early-morning transition period and also help to quantify the onset, depth, intensity, etc. of valley flows. These observations

may also be used to help validate mesoscale models used by the NWS including e.g. the High-Resolution Rapid Refresh (HRRR) and the Weather Research and Forecasting (WRF) models, as well as dispersion models that use these fields in their initialization (e.g., NOAA's Hybrid Single-Particle Lagrangian Integrated Trajectory (HYSPLIT) model. Furthermore, these sUxS observations may also be used to improve numerical weather prediction models (e.g., Chilson et al. 2019; Leuenberger et al. 2020; McFarquhar et al. 2020) by assisting in the development of better surface layer and ABL parameterization

schemes (e.g., Lee and Buban 2020, Lee et al. 2021).

## 2 Methods

### 2.1. Site Description

All sUxS flights in this study were conducted at the Oliver Springs Airport (36.038 N, 84.308 W, 240 m above mean sea level (MSL)), which is located approximately 35 km northwest of Knoxville, Tennessee (Figure 1a). The airport at Oliver

Springs is sparsely used for full-scale aircraft and includes a grassy runway with approximate dimensions of 850 m × 60 m





(Figure 1b). The region surrounding the Oliver Springs Airport is characterized by forested hillslopes, and the eastern edge of the Cumberland Plateau lies about 20 km to the north and west of the site. Cropland is the more dominant landuse type immediately to the south and east of the site.

### 2.2. Platform / sensor payload and sensor validation

All sUxS flights were conducted using the Meteomatics Meteodrone SSE, owned by the National Oceanic and Atmospheric Administration (NOAA) Air Resources Laboratory (ARL) Atmospheric Turbulence and Diffusion Division (ATDD). The Meteodrone SSE is powered by six electric motors, has a wingspan of approximately 0.4 m, and a gross weight of 0.7 kg. The Meteodrone uses a bead thermistor to measure temperature and a capacitive sensor for relative humidity. The thermistor and capacitive sensor have manufacturer-stated accuracies of $\pm$ 0.1 °C and < 2 %, respectively, and response time of < 1 s

and < 4 s, respectively (Table 1). Prior to being deployed on the sUxS and following the procedure described by Lee et al. (2019), the bead thermistor and capacitive relative humidity sensor were calibrated in NOAA / ARL / ATDD's National Institutes for Standards and Technology (NIST)-traceable Thunder Scientific model 2500 two-pressure humidity generator. We used three temperature set points (10 °C, 20 °C, and 30 °C) and five relative humidity set points (20%, 40%, 60%, 80%, and 94%) for 15 total temperature / relative humidity combinations, following previous studies that evaluated temperature /

humidity sensors used on board sUxS (e.g. Lee et al. 2019). The tests in the NIST chamber indicated that the Meteodrone's temperature sensor has a cold bias of up to 0.3 °C at low temperature / relative humidity combinations. Additionally, the Meteodrone's relative humidity sensor has a dry bias ranging from 2% for low relative humidity up to about 9% for relative humidity >80%. The dry bias is corroborated by findings from Koch et al. (2018) who compared measurements from the Meteodrone's sensors with those from rawinsondes. Koch et al. found that the Meteodrone has a dry bias of 7%, but

conversely found a warm bias in temperature of about 0.4 °C.

In addition to having sensors to measure temperature and relative humidity, the Meteodrone has a piezo resistive pressure sensor with a manufacturer-stated accuracy of ±0.1 mb (Table 1). Additionally, wind speed and wind direction are computed using the Meteodrone's onboard inertial sensors and global positioning system (GPS). The manufacturer-stated GPS

horizontal position uncertainty is 1 m s$^{-1}$, and the manufacturer-stated compass uncertainty is < 10°. The response time of the data is < 4 Hz.

Although the manufacturer does not give an explicit estimate of the errors in the wind speed and wind direction, Koch et al. (2018) evaluated the Meteodrone against rawinsonde observations and found a positive wind speed bias of 0.2 m s-1 and

clockwise bias in wind direction of 7° which provide additional confidence in the fidelity of the measurements from the Meteodrone.



## 2.3. Flight strategy

On 29 of the 31 days, we conducted flights on eight times per day; three and six flights were conducted on 9 September and 10 December, respectively. On 12 days between 20 August and 8 September 2020, the sUxS profiles commenced within 5 min of local sunrise (Table 2). On the remaining days between 16 September and 10 December 2020, the first sUxS profile began about 30 min following local sunrise. We used ARL's certificate of authorization (COA) from the Federal Aviation Administration (FAA), which allowed us to perform sUxS flights up to 1000 m AGL. For all sUxS flights, constant 3 m s$^{-1}$ ascent and descent rates were used, and thus the average flight time was about 10 min.

ARL's COA with the FAA stipulates that the sUxS must remain within visible line of sight to the unaided eye at all times. For this reason, the maximum altitude to which the sUxS was flown was typically >650 m AGL, which occurred during 77% of all flights (Figure 2). During the remaining times, fog and low-level clouds at the site obscured on-site visibility and prevented the sUxS from reaching this altitude.

## 2.4. Data processing

All raw meteorological data were sampled at 10 Hz and stored onboard the aircraft. Only data collected during the sUxS's ascent were used in all post-processing. Following each flight, any value that was recorded as "NA" was changed to -9999.00. Next, we ensured that each measurement did not fall outside a physically-possible range. To this end, we flagged as -9999.00 any temperatures <-40°C or >50°C, relative humidity <0% or >100%, dew point temperatures <-40°C or >50°C, air pressures <10 mb or >1050 mb, wind speeds < 0 kt or >100 kt, wind directions < 0° or >360°, aircraft altitudes < 0 m or > 10000 m, and latitude or longitudes >5° from Oliver Springs' coordinates to remove instances in which there was an incorrect GPS fix. If there are any flags for a given altitude, all values at that altitude were discarded and were not used for further post-processing. Afterwards, profiles of temperature, dew point, pressure, wind speed, and wind direction were plotted and visually inspected to ensure all erroneous values had been successfully discarded and that all values were within a physically-possible range.

Output files from the Meteodrone were stored as .CSV files prior to being converted into .NSP files. The .NSP format enables the data to be ingested into the NWS's Decision Support Service (DSS) Advanced Weather Interactive Processing System (AWIPS) systems, which is a tool forecasters use for visualizing meteorological data. Additionally, time-height cross-sections of measured and derived meteorological variables were generated and stored as .PNG files, as shown in the example in Figure 3 in which temperature, relative humidity, dew point, and wind speed are contoured as a function of height and time on 8 September 2020.





## 3. Examples of collected sUxS profiles

### 3.1. 1-4 September 2020

Figure 4 shows the temporal evolution of potential temperature, specific humidity, wind speed, and wind direction over four
consecutive days in early September in which sUxS flights were conducted eight times daily. Although the majority of flights on these days exceeded 500 m AGL, flights during the early morning on 1 September were only flown to a maximum altitude of 200-300 m AGL because of low-level fog in the area obscuring visibility of the sUxS. Overall, the flights conducted every 30 min provided finescale details on the evolution of the low-level thermodynamic and kinematic structure on these respective days. For example, with these measurements we are able to observe the transition from near-surface
stable conditions to a near-surface super-adiabatic layer that formed during the mid-late morning hours. We also observed near-surface drainage flows during the early morning from the east and east-southeast that extend up to about 100-200 m AGL. During the morning transition, these winds veered to the south, consistent with the synoptic flow on these days.

### 3.2. 1-4 September 2020

Similarly, Figure 5 illustrates the temporal evolution in potential temperature, specific humidity, wind speed, and wind
direction, but over four consecutive days from 30 November through 3 December. A cold front passed through the region around 0900 UTC on 30 November, resulting in cold air advection during the early-morning hours on 30 November and northwesterly flow throughout the depth of the sUxS profile on both 30 November and 1 December. On 2 and 3 December, however, more classical ABL evolutions were observed, including, for example, evidence of easterly drainage flows during the early morning on 3 December.

**4 Data availability and file structure**

The datasets are available from the NOAA / ARL /ATDD's anonymous ftp server, which can accessed at <ftp://ftp.atdd.noaa.gov/pub/mrx/suas>. The filenames have the convention ATDD_sUAS_YYYYMMDD_HHMM.nsp, where YYYY, MM, DD, HH, and MM correspond with the year, month, day, hour, and minute, respectively, at which the sUxS profile began. Each .NSP file contains the pressure (units: mb), height above mean sea level (units: m), temperature
(units: °C), dew point temperature (units: °C), wind direction (units: °), and wind speed (units: kt).

**5 Summary and outlook**

We performed 241 sUxS flights over 31 days between 20 August and 10 December 2020 near Oliver Springs, Tennessee to sample the evolution of temperature, moisture, pressure, wind speed, and wind direction fields during the early morning transition period over complex terrain in eastern Tennessee. Besides being provided in real-time to the nearest NWS WFO in



Morristown, Tennessee to support weather forecasting operational needs there, these datasets provided finescale information on the evolution of the ABL. Having this information over nearly three dozen days across different seasons helps to support the initialization of high-resolution mesoscale models such as the HRRR and WRF, as well as HYSPLIT, and can also be used to investigate ABL transport and mixing processes occurring over complex terrain.

**Author Contributions**

All authors contributed to the generation of this dataset. TRL, ED, MB, TPM, and CBB contributed to the experimental design; TJS and ED conducted the sUxS flights; TJS and ED assisted with the data post-processing; and TRL, TJS, MB, ED, TPM, and CBB prepared the manuscript.

**Competing Interests**

The authors declare no competing interests.

**Acknowledgements**

We thank Mr. John Williams of the Oliver Springs Airport for allowing us to use the facility for our sUxS profiles. We also

thank Dr. Lukas Hammerschmidt and Dr. Martin Fengler of Meteomatics for assisting with the post-processing of the Meteodrone's datasets and who provided suggestions on an earlier version of this manuscript. Lastly, we note that the results and conclusions of this study, as well as any views expressed herein, are those of the authors and do not necessarily reflect those of NOAA or the Department of Commerce.

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



**Table 1. Manufacturer-stated specifications of the sensors on board the Meteomatics Meteodrone SSE.**

| Variable | Type | Range | Response Time | Accuracy |
|---|---|---|---|---|
| Temperature | Bead Thermistor | -95°C to +50°C | < 1 s | ± 0.1 °C |
| Relative humidity | Capacitive | 0-100% RH | < 4 s | < 2% RH |
| Pressure | Piezo resistive | 10-1200 hPa | 250 ms | ±0.1 hPa |
| Wind speed | A/C flight path | 0-40 m/s | < 250 ms | < 1 m s$^{-1}$ |
| Wind direction | A/C flight path | 0-360° | < 250 ms | < 10° |

**Table 2. Date, time of local sunrise, time of first sUxS profile, time of last sUxS profile, and minutes of first profile from local sunrise.**

| Date in 2020 | Local sunrise (UTC) | First sUxS profile (UTC) | Last sUxS profile (UTC) | Minutes of first profile from local sunrise |
|---|---|---|---|---|
| 20 Aug | 1100 | 1104 | 1437 | 4 |
| 21 Aug | 1100 | 1105 | 1434 | 5 |
| 24 Aug | 1103 | 1106 | 1434 | 3 |
| 25 Aug | 1103 | 1107 | 1435 | 4 |
| 26 Aug | 1104 | 1104 | 1432 | 0 |
| 27 Aug | 1105 | 1105 | 1434 | 0 |
| 31 Aug | 1108 | 1103 | 1434 | -5 |
| 1 Sep | 1109 | 1105 | 1435 | -4 |
| 2 Sep | 1110 | 1101 | 1433 | -9 |
| 3 Sep | 1110 | 1104 | 1434 | -6 |
| 4 Sep | 1111 | 1104 | 1434 | -7 |
| 8 Sep | 1115 | 1120 | 1450 | 5 |
| 9 Sep | 1115 | 1124 | 1319 | 9 |
| 16 Sep | 1120 | 1150 | 1520 | 30 |
| 23 Sep | 1126 | 1205 | 1535 | 39 |
| 30 Sep | 1131 | 1204 | 1534 | 33 |
| 7 Oct | 1137 | 1201 | 1535 | 24 |
| 13 Nov | 1212 | 1249 | 1619 | 37 |
| 16 Nov | 1215 | 1249 | 1619 | 34 |
| 17 Nov | 1216 | 1249 | 1619 | 33 |
| 18 Nov | 1217 | 1249 | 1619 | 32 |
| 24 Nov | 1223 | 1250 | 1619 | 27 |
| 25 Nov | 1224 | 1249 | 1619 | 25 |
| 30 Nov | 1228 | 1304 | 1633 | 34 |
| 1 Dec | 1229 | 1305 | 1634 | 36 |
| 2 Dec | 1230 | 1304 | 1634 | 34 |
| 3 Dec | 1231 | 1304 | 1634 | 33 |
| 7 Dec | 1235 | 1304 | 1634 | 29 |
| 8 Dec | 1235 | 1304 | 1634 | 29 |
| 9 Dec | 1236 | 1304 | 1634 | 28 |
| 10 Dec | 1237 | 1304 | 1534 | 27 |





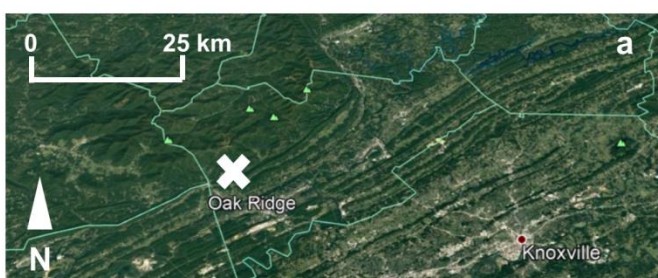

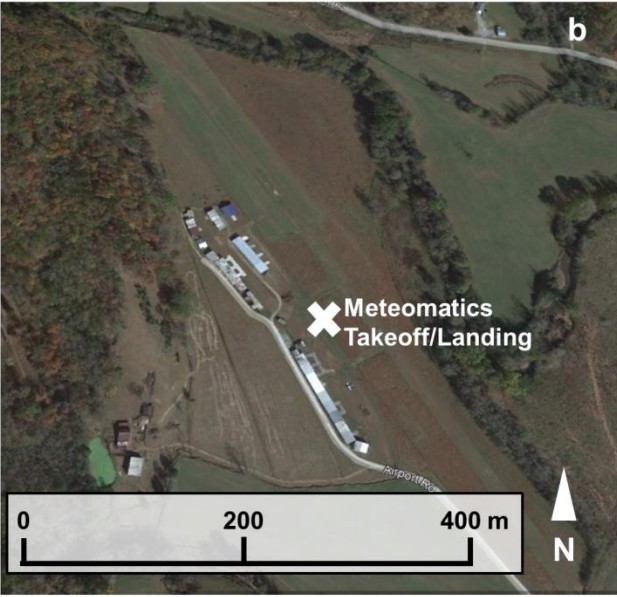

**Figure 1: (a) Location of Oliver Springs Airport (white X) relative to Oak Ridge, TN and Knoxville, TN; and (b) land surface**
**surrounding sUxS profiling site (white X) at Oliver Springs. Images courtesy of Google Earth, © Google Earth.**



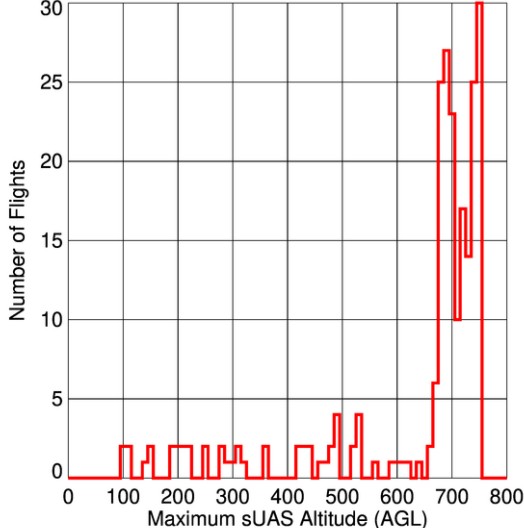

**Figure 2: Distribution of the maximum number sUxS altitude for all sUxS flights conducted with the Meteomatics Meteodrone at the Oliver Springs Airport. Data binned every 10 m.**


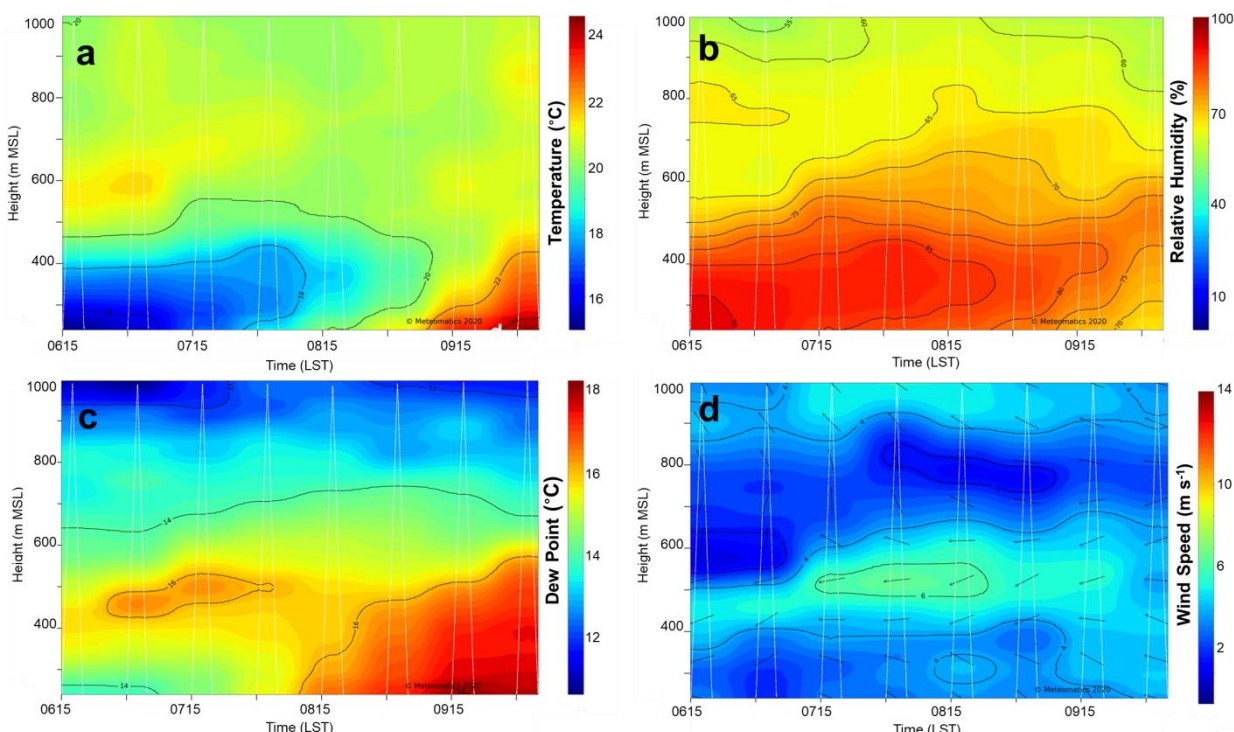

**Figure 3: Time-height cross-sections of (a) temperature, (b) relative humidity, and (c) dew point on 8 September 2020. Panel (d) shows a time-height cross-section of wind speed with wind direction (arrows) overlaid. These example plots were generated in near-real time and provided to the local NWS WFO. Time is in LST (UTC-5). White lines indicate the flight profiles with respect to time of each sUAS flight.**




**Figure 4: (a) sUxS-derived potential temperature as a function of time of day on 1 September 2020. Same for (b), (c), and (d), but for 2, 3, and 4 September 2020, respectively. Same for panels (e-h), (i-l), and (m-p) but for specific humidity, wind speed, and wind direction, respectively.**

**Figure 5: (a) sUxS-derived potential temperature as a function of time of day on 30 November 2020. Same for (b), (c), and (d), but for 1, 2, and 3 December 2020, respectively. Same for panels (e-h), (i-l), and (m-p) but for specific humidity, wind speed, and wind direction, respectively.**