# Peer review of "NOAA Air Resources Laboratory Atmospheric Turbulence and Diffusion Division's Measurements of Temperature, Humidity and Wind using Small Uncrewed Aircraft Systems to Support Short-Term Weather Forecasting Needs over Complex Terrain"

_Earth System Science Data, 2022_

## Referee Comment (RC1)

The authors provide a nice summary of UAS observations that were collected during the transition from summer to winter of 2020 near Oliver Springs, TN. The authors do a nice job of summarizing the data collected and provide a couple of case examples, but fail to really connect how the data were used to support short-term weather forecasting. If the authors could tie in a bit more into how the observations might have been used by the short-term or aviation desk at the Knoxville WFO, it would better tie into the purpose of the study and the title for that matter. For example, perhaps the short term desk used the wind observations to improve their wind forecasts at local airports or perhaps the data collected on another day might have been used to enhance predictions of the timing of fog burn off. A brief description of how the data were used and any feedback from the weather forecast office would be valuable.

Major comments:
1) Aside from the discussion above, my only other major comment is that the authors should obtain a DOI for their dataset and also perhaps mention that it will be added to periodically as I see that some new observations have been added from Dec 2021.
2) There are a couple of suggestions for new figures that I highly recommend, however, these are not really major items.

Some specific comments:

Line 1: consider shortening the title. For example, perhaps listing variable measured in the title is not necessary.

Line 12: I applaud the use of the gender neutral term Uncrewed Aircraft Systems, however, the authors introduce a new acronym (UxS) that would seem to generate confusion. It is not clear where the 'x' comes from or what it means and the use of this new acronym is confusing since most other recent papers in the literature are using UAS as the acronym.

Line 16: How far is the UAS profiling site and Knoxville forecast area from Nashville (BNA) WFO where radiosondes are launched. This is a key aspect of UAS ability to collect gap filling measurements in both time and space.

Line 21: I feel that it is important that the authors obtain a DOI for the dataset. This can be easily obtained using NOAA's climate data server: https://www.ncdc.noaa.gov/cdo-web/ or some other online data archiving service.

Line 53: may also be used to improve analyses needed by NWP models through data assimilation (e.g., Chilson et al. 2019; Leuenberger et al. 2020; McFarquhar et al. 2020) and by assisting …..

I suggest adding the recent paper published by Jensen et al. who demonstrated the potential utility of targeted UAS observations in improving the analysis and prediction of terrain-drive flows. This is pretty relevant to the topics presented in this study with respect to using gap filling UAS to detect local flow patterns.

Jensen, A. A., J.O. Pinto and Coauthors, 2021: Assimilation of a coordinated fleet of UAS observations in complex terrain: EnKF system design and preliminary assessment, *Mon. Wea. Rev.*, 149, 1459-1480, DOI: 10.1175/MWR-D-20-0359.1

Line 59: while Figure 1 gives some perspective on the variability of land surface type surround the profiling site, it is difficult to make out topographic variability. Perhaps a top map or terrain contour map can be added. This would help the ready better appreciate the terrain features and how they related to representativeness of the measurements.

Line 65: Add a reference describing the Meteodrone.

Line 70: Are the measurements of RH corrected to account for the 4 s lag in sensor response time? Assuming a 3 m s-1 vertical rate, the temperature and RH measurements can be "displaced" by roughly 10 m which can have an impact on estimates of water vapor mixing ratio where vertical gradients are large.

Line 80: Has measurement error been quantified during profiling maneuvers?  Often the biases can depend on whether the UAS is ascending or descending due to prop-wash effects. Also, is a radiation shield or aspiration used?

Line 85: Why is horizontal position error given in meters/sec?

Line 105: Why were only ascent profile data used?

Line 110: It seems like the error bounds for min air pressure, max wind, and lat and lon position errors are under-restrictive. For example, while the NWS users may have been aware of where the data were collected, the data would not be very useful in data assimilation and other modeling studies if the position error is more than 100 m.

At the very least the position information should be included within the data files or in a README file that also include the reference to this paper. Other attributes such as sensor type and UAS model could also be included.

Line 126: It would very useful to add a figure showing the time-height cross section or profiles of relative humidity for 1 Sept to demonstrate the potential utility of UAS observations in the short term prediction of fog and its evolution.

Line 134: Heading should be 30 Nov – 3 December.

Line 150: change "helps" to "could help"

Line 254: The table provides a nice summary of the flights, but perhaps the last column could be replaced with number of profiles flown and a column could be added that provide maximum altitude attained.

Line 270: There is a bit of a kink in the potential temperature profiles just above the ground  that is most apparent in figures c & d. Is there any explanation for what might be causing this?